# Molecular Mechanisms of Resistance to Ionizing Radiation in *S. cerevisiae* and Its Relationship with Aging, Oxidative Stress, and Antioxidant Activity

**DOI:** 10.3390/antiox12091690

**Published:** 2023-08-30

**Authors:** Alejandro González-Vidal, Silvia Mercado-Sáenz, Antonio M. Burgos-Molina, Juan C. Alamilla-Presuel, Miguel Alcaraz, Francisco Sendra-Portero, Miguel J. Ruiz-Gómez

**Affiliations:** 1Departamento de Radiología y Medicina Física, Facultad de Medicina, Universidad de Málaga, 29010 Málaga, Spain; alegvidal1993@gmail.com (A.G.-V.); alamillajc@hotmail.com (J.C.A.-P.); sendra@uma.es (F.S.-P.); 2Instituto de Investigación Biomédica de Málaga y Plataforma en Nanomedicina (IBIMA Plataforma BIONAND), Parque Tecnológico de Andalucía (PTA), 29590 Málaga, Spain; smercad@uma.es (S.M.-S.); aburgos@uma.es (A.M.B.-M.); 3Departamento de Fisiología Humana, Histología Humana, Anatomía Patológica y Educación Físico Deportiva, Facultad de Medicina, Universidad de Málaga, 29010 Málaga, Spain; 4Departamento de Especialidades Quirúrgicas, Bioquímica e Inmunología, Facultad de Medicina, Universidad de Málaga, 29010 Málaga, Spain; 5Departamento de Radiología y Medicina Física, Facultad de Medicina, Universidad de Murcia, 30100 Murcia, Spain; mab@um.es

**Keywords:** ionizing radiation, aging, oxidative stress, free radicals, reactive oxygen species, ROS, antioxidant activity, yeast, *S. cerevisiae*

## Abstract

The repair of the damage produced to the genome and proteome by the action of ionizing radiation, oxidizing agents, and during aging is important to maintain cellular homeostasis. Many of the metabolic pathways influence multiple processes. In this way, this work aims to study the relationship between resistance/response to ionizing radiation, cellular aging, and the response mechanisms to oxidative stress, free radicals, reactive oxygen species (ROS), and antioxidant activity in the yeast *S. cerevisiae*. Systems biology allows us to use tools that reveal the molecular mechanisms common to different cellular response phenomena. The results found indicate that homologous recombination, non-homologous end joining, and base excision repair pathways are the most important common processes necessary to maintain cellular homeostasis. The metabolic routes of longevity regulation are those that jointly contribute to the three phenomena studied. This study proposes eleven common biomarkers for response/resistance to ionizing radiation and aging (EXO1, MEC1, MRE11, RAD27, RAD50, RAD51, RAD52, RAD55, RAD9, SGS1, YKU70) and two biomarkers for response/resistance to radiation and oxidative stress, free radicals, ROS, and antioxidant activity (NTG1, OGG1). In addition, it is important to highlight that the HSP104 protein could be a good biomarker common to the three phenomena studied.

## 1. Introduction

Ionizing radiation produces different types of DNA damage, with double-strand breaks (DSB) being especially important. In this sense, the repair capacity contributes not only to the maintenance of the genome integrity but also to the resistance to radiation [1,2,3]. In addition, the response to radiation to ensure cell homeostasis activates other mechanisms involved in cell cycle blockage, free radicals’ formation, and apoptosis inhibition [1].

DSB can also be produced by products generated during cellular metabolism, occurring mainly during replication. Hydrolysis, oxidation, and non-enzymatic methylation phenomena can produce DNA modifications that contribute to the appearance of DSB [4]. The cellular state, young or old, the metabolic state, the phase of the cell cycle, etc., contribute to the ability to repair the induced damage.

Spontaneous or metabolic-induced lesions that appear during cell replication are repaired predominantly by homologous recombination (HR) due to the proximity of the two sister strands. However, the breaks induced by external agents that appear in distant places between homologous sequences are repaired by non-homologous end-joining (NHEJ) [5,6]. HR is free of errors, but NHEJ frequently produces errors. It is preferred to induce an error rather than not repair the damage, in order to guarantee the stability of the genome and cell viability [7]. In *S. cerevisiae*, the role of HR in DSB repair predominates [8,9].

Mutations that appear in the DSB repair genes can trigger processes of radiosensitivity, premature aging, immunodeficiency, and carcinogenesis [7]. The appearance of reactive oxygen species (ROS) after exposure to DNA-damaging agents (such as ionizing radiation) activates the signaling pathways that repair the induced damage. Chronic exposure to ROS is associated with the induction of cancer and with the development and progression of neurodegenerative and cardiovascular diseases. A connection between ROS exposure and the phenomenon of aging has also been described [10].

The aging phenomenon is a multifactorial process in which a high energy metabolism is essential to maintain cellular homeostasis, repair damage, and slow down the process of cellular deterioration. Thus, the decline in energy metabolism caused by DNA-damaging agents accelerates the aging process and, conversely, high glycolytic and respiratory activity increase the cellular resistance to environmental agents that cause stress [11].

Recent studies show that not only are the damages caused to the DNA molecule important, but so are those produced on the proteins. In this sense, the response of proteins to stress caused by radiation also activates molecular response mechanisms, especially chaperones. Heat shock proteins (HSP) play an important role in the maintenance of proteostasis through the correct folding of damaged proteins [12]. However, when misfolding proteins are oxidized, the damage is fixed and therefore chaperones cannot restore folded proteins correctly. Cellular aging also produces the appearance of proteins damaged by oxidation, which accumulate with age, causing the loss of cellular homeostasis. Therefore, there is a connection between aging, damage caused by ROS, and radiation. This connection can be at the genomic or proteomic level or at both levels at the same time [13].

The mechanisms of response to ionizing radiation and to agents causing oxidative damage, as well as to damage and alterations in the proteome during cellular aging, are highly dynamic processes. During the peak period of cellular response, the number and type of active molecules is more important than during the initiation and attenuation stages [14]. This overexpression of biomolecules highlights the multiple roles that many of them play together in the processes of radiation response, aging, and response to oxidative stress.

The yeast *Saccharomyces cerevisiae* constitutes a model organism whose gene characteristics and functions are well described in multiple curated databases. Systems biology and the use of bioinformatics tools allows us to address the need to know the common molecular mechanisms between different phenomena, to study their genetic basis, and their regulation, with the ultimate need being to search for molecular markers common to different phenomena [15].

The aim of this work is to study the relationship between resistance/response to ionizing radiation, cellular aging, and the response mechanisms to oxidative stress, free radicals, reactive oxygen species (ROS), and antioxidant activity in the yeast *S. cerevisiae*, in order to understand the mechanisms of interaction and regulation between them and to search for molecular markers related to these phenomena.

## 2. Materials and Methods

### 2.1. Study Strategy and Analysis

The search and analysis strategy in this work was divided into three stages (Figure 1). Firstly, a search for common genes that relate the phenomenon of resistance to ionizing radiation, cellular aging, and oxidative response was carried out. For text/data mining, the GenAge database, STRING database/search engine, and the pubmed2ensembl search engine were used. Subsequently, a second stage was carried out where an enrichment analysis of the common genes found was performed. This stage was divided into a functional analysis that consisted of a gene ontology (GO) analysis, Kyoto Encyclopaedia of Genes and Genomes (KEGG) pathway analysis, and an enriched ontology clusters identification [16]. Finally, in stage 3, the identification of the main genes involved in resistance to radiation and aging was carried out, as well as the analysis of protein–protein interactions and the study of regulatory mechanisms [15].

### 2.2. Text/Data Mining

The most complete and updated data source was used to search for specific genes of each biological process. Data mining in the specific GenAge database (https://genomics.senescence.info/genes/index.html) (accessed on 24 February 2023) allowed us to search for all validated genes of *Saccharomyces cerevisiae* involved in the aging process. This database does not contain senescence genes, which makes it possible to discriminate the genes related to the phenomenon of aging with respect to those involved in cellular senescence. All genes involved in the aging process related to pro-longevity, anti-longevity, and those necessary for fitness were downloaded. 

The search for genes involved in the oxidative response did not show results using the pubmed2ensembl search engine. Therefore, the Search Tool for the Retrieval of Interacting Genes (STRING) database (https://string-db.org/) (11.5 version) (accessed on 15 March 2023) was used to identify *S. cerevisiae* genes related to oxidative stress, free radicals, antioxidant activity, and ROS. This database/search engine allowed us to obtain results from both the literature and curated databases.

Since there is no specific database of *S. cerevisiae* genes/proteins for resistance or response to ionizing radiation, the text mining for them was carried out using the pubmed2ensembl search engine (http://pubmed2ensembl.ls.manchester.ac.uk/) (accessed on 24 January 2023), which allows links between the literature and genes related to different processes [15,17]. Only the genes that appear in the literature related to *S. cerevisiae* were searched using two queries with the terms “ionizing AND radiation AND resistance NOT ultraviolet NOT UV” and “ionizing AND radiation AND response”. Search parameters were set to “search for PubMed IDs”, “retrieve up to 100,000 document IDs”, and “filter on MEDLINE PubMed ID”. 

Numerous authors have described the genes that participate in the response to radiation, also referring to the fact that they participate in resistance phenomena. The criteria used to define radiation resistance were the participation of genes/proteins in processes described by the authors that are correlated with the following:DNA repair mechanisms (DSB repair, HR, NHEJ, base excision repair, mismatch repair, etc.).DNA-damage checkpoint control proteins (mitosis entry checkpoint, telomere length regulation, etc.).Cell cycle division control (G1/S-specific cyclins, cell-cycle box factors, and regulatory proteins in response to radiation).Heat shock response.Proteins that relate cell proliferation with resistance to radiation.Regulatory proteins and post-replication repair ubiquitin-proteins in response to radiation.General transcription and DNA repair factors.Other processes related to radiation response (proteins involved in joint resistance to radiation and metals and/or drugs, helicases, and transcriptional coactivators).

A list of genes was obtained after removing duplicates. Subsequently, all searches were compared to obtain the list of genes common to each phenomenon. Common genes were used for further analysis.

### 2.3. Analysis of Gene–Gene Interaction Networks and Clusters Identification

After the identification of the genes involved in each process, an analysis of interactions was carried out in order to identify the molecular pathways and all clusters involved in the resistance and response to radiation as well as in the aging process. At this stage, all genes found were considered, and an independent analysis was performed for each group of genes. Metascape (http://metascape.org) (accessed on 29 March 2023) [18] was used to investigate all biological processes and clusters. *S. cerevisiae* was selected as the target organism. GO enrichment analysis was applied to each MCODE network to extract “biological meanings” from the network component, where the top three best *p*-value terms were retained.

### 2.4. Gene Ontology and KEGG Pathway Analysis

The selected common genes were used to perform a functional enrichment analysis using GeneCodis (http://genecodis.cnb.csic.es) (accessed on 3 April 2023) [19,20]. This bioinformatics tool returns information related to GO biological processes, GO cellular components, and GO molecular functions. An additional functional analysis was performed using the KEGG pathway enrichment analysis in GeneCodis (https://www.genome.jp/kegg/) (accessed on 5 April 2023) [21,22]. *S. cerevisiae* was adjusted as a model organism. The hypergeometric statistical test was chosen. The *p*-value was adjusted to obtain the most enriched GO and KEGG annotations [23,24,25,26].

### 2.5. Enriched Ontology Clusters

Clusters identification was performed using Metascape [18]. *S. cerevisiae* was selected in the input as species and in the analysis as species. The enrichment analysis was carried out following the ontology sources obtained in GO biological processes, KEGG pathways, Reactome gene sets, Wikipathways, and PANTHER pathway. All genes in the genome were used as the enrichment background. Terms with a *p*-value < 0.01 (based on cumulative hypergeometric distribution), a minimum count of 3, and an enrichment factor >1.5 were collected and grouped into clusters based on their membership similarities. Kappa scores was used as the similarity metric when performing hierarchical clustering on the enriched terms, and sub-trees with a similarity of >0.3 were considered a cluster. The most statistically significant term within a cluster was chosen to represent the cluster. The molecular complex detection (MCODE) algorithm, supplied by Metascape, was used to identify neighborhoods where proteins are densely connected allowing the identification of its components [27]. Then, each MCODE components was submitted to independent enrichment investigation, interaction analysis, and regulation study.

### 2.6. Protein–Protein Interaction Analysis

The STRING database (11.5 version), an online platform of protein–protein interaction networks and functional enrichment analyses [2,28], was used to analyse the main interactive relationships among the proteins found in the MCODE cluster. The analysis was categorized according to local network clusters (STRING), KEGG pathways, and REACTOME (https://reactome.org/) (accessed on 18 April 2023). Edges in the network represent protein–protein associations and each node represent all proteins produced by a single protein-coding gene locus. The minimum interaction score for edge confidence was 0.400. Physical subnetworks were obtained where edges indicate that the proteins are part of a physical complex and the line thickness in the network indicate the strength of data support (confidence) [27].

### 2.7. Regulation Mechanisms

The regulatory mechanisms of the MCODE cluster proteins were obtained from the *Saccharomyces* Genome Database (https://www.yeastgenome.org/) (accessed on 21 April 2023). The regulatory networks, which include the regulator, target, and focus, were obtained for each gene.

### 2.8. Human, Mouse, and Rat Homologous Genes and Their Comparison with More Specific Databases

The search for human (*Homo sapiens*), mouse (*Mus musculus*), and rat (*Rattus norvegicus*) homologous genes, starting from *S. cerevisiae*, was performed in the “HomoloGene” database (NCBI: National Center for Biotechnology Information) (https://www.ncbi.nlm.nih.gov/homologene/?term) (accessed on 16 August 2023) and in the GenAge database. Homologous genes were compared in the RadBioBase (http://radbiodb.physics.ntua.gr/) (accessed on 22 August 2023) database, specific for evaluating the effect of ionizing radiation in human, mouse, and rat, and in the MGI (Mouse Genome Informatics) database (https://www.informatics.jax.org/) (accessed on 22 August 2023), specific to the type of stress studied.

## 3. Results

### 3.1. Text/Data Mining

The results obtained through the search strategy are shown in Figure 2. The search performed in Pubmed2ensembl showed 26 and 46 genes related to the terms “ionizing AND radiation AND resistance” and “ionizing AND radiation AND response”, respectively. The GenAge database showed 911 genes related to aging in *S. cerevisiae*; 51 pro-longevity, 279 anti-longevity, and 492 necessaries for fitness. The STRING search revealed 139 genes related to oxidative stress, 20 to free radicals, 41 to ROS, and 35 to antioxidant activity in *S. cerevisiae*.

After eliminating the duplicated genes and comparing the different groups, 27 common genes between the response to radiation and cell aging were obtained. These genes were *CAT5*, *CDC6*, *DOT1*, *EXO1*, *HSP104*, *MEC1*, *MRE11*, *MSH2*, *NTH1*, *RAD10*, *RAD27*, *RAD34*, *RAD4*, *RAD50*, *RAD51*, *RAD52*, *RAD55*, *RAD57*, *RAD9*, *RAS2*, *RCK2*, *SAS3*, *SGS1*, *SIR2*, *SWI6*, *VPS8*, and *YKU70*. Between the response/resistance to radiation and the group related to antioxidant defense and oxidative stress, there were only four common genes (*NTG1*, *SIT4*, *OGG1*, *HSP104*). However, it is important to highlight that only the *HSP104* gene was common to the three groups (Table 1).

### 3.2. Analysis of Gene–Gene Interaction Networks and Clusters Identification

The analysis of interactions for the search terms “ionizing AND radiation AND resistance NOT ultraviolet NOT UV” (Figure 3A) showed a network of interactions where 11 statistically enriched terms were identified (GO/KEGG terms, canonical pathways, hall mark gene sets, etc.). Among them, the group of proteins involved in DNA repair stands out for its highest statistical probability, followed by those related to homologous recombination (*p* < 1 × 10^−20^). It is important to highlight the regulation cluster of TORC1 signalling. The search terms “ionizing AND radiation AND response” showed a network of interactions (Figure 3B) where 20 statistically significant processes were identified. Those corresponding to DNA repair as well as DNA integrity checkpoint signalling (*p* < 1 × 10^−20^) stood out due to their higher probability. They also highlighted the cluster related to telomere maintenance, DNA metabolic process, and the regulation of the G1/S transition of the mitotic cell cycle. Figure 3C shows the interactions of aging genes. Twenty processes and numerous clusters were identified, some of them independent. The proteins related to the ribosome, the longevity regulation pathways, the phenomenon of microautophagy of the nucleus, and various metabolic processes stand out with a greater probability. Figure 3D shows the network obtained for the 139 genes described with the term “antioxidant”. Twenty processes were observed where the cellular response to oxidative stress (*p* < 1 × 10^−100^) and cellular oxidant detoxification (*p* < 1.78 × 10^−35^) stood out as more significant along with other processes such as cell redox homeostasis, cellular response to heat, glutathione metabolism, and apoptosis. The twenty genes obtained for the “free radicals” concept showed a network with four processes whose highest probability was for the autophagy process (*p* < 5.5 × 10^−18^) (Figure 3E). The term “antioxidant” showed a network with thirty-five genes (Figure 3F), observing interactions of nine processes whose highest probabilities were for cellular oxidant detoxification (*p* < 3.3 × 10^−72^), a response to oxidative stress (*p* < 2.5 × 10^−41^), a reactive oxygen species metabolic process (*p* < 4.5 × 10^−25^), and cell redox homeostasis (*p* < 3.6 × 10^−21^), among others. Finally, the network obtained for the 41 genes of the term “reactive oxygen species” showed 13 processes, dominated by the response to reactive oxygen species (*p* < 7.9 × 10^−41^) and the detoxification of reactive oxygen species (*p* < 2.2 × 10^−15^). The response to heat and cell redox homeostasis also intervened in the network, although with less probability (Figure 3G).

The networks obtained showed the processes involved in each of them, ordered by their statistical significance (hypergeometric test). Within each network, each process constitutes a cluster where the proteins involved interact with each other. Figure 3 shows the networks, interactions, and processes separately for each of the search terms used. Many of the processes were repeated in several networks, revealing the interaction between many of these proteins, therefore suggesting a common basis between these phenomena. From this point, it is important to study the common genes in these networks and their interactions, so that with the help of an enrichment analysis, the common processes and genes between them can be obtained.

### 3.3. Gene Ontology and KEGG Pathway Analysis

The 27 common genes for radiation resistance and cellular aging were analysed to assess GO biological process, GO cellular component, GO molecular function, and KEEG pathway analysis, using GeneCodis. This application performs an enrichment of the terms that statistically show greater significance and shows the genes involved in each process. The genes analysed were involved in 163 biological processes. However, only 98 annotations were significant with an adjusted *p*-value < 0.05. To enrich the process and select the most relevant genes and biological processes involved in resistance/response to radiation and aging, a much smaller adjusted p threshold was used, establishing the cut off in the annotations of biological processes with (*p* < 0.0001, hypergeometric test). With this cut-off level, 17 biological processes related to both phenomena and 25 genes involved were found. The most enriched biological processes are shown in Figure 4A. In this sense, the most enriched processes were mitochondrial double-strand break repair via homologous recombination and gene conversion at a mating-type locus. DNA repair and cellular response to DNA damage stimulus showed the highest statistical significance and a greater number of genes. They were followed by the processes of DNA recombination, telomere maintenance, etc. (Figure 4A). The GO cellular component analysis showed that most of the processes involved were found in the cell nucleus. It should be noted that some of them were also located in the XPC complex, Rhp55-Rhp57 complex, and nuclear chromosome, among others (*p* < 0.005, hypergeometric test). The analysis of the molecular functions indicated that the largest number of genes were involved in protein binding and DNA binding. Double-stranded DNA binding and DNA-binding functions showed the highest probability. It is worth noting the fact that the DNA strand exchange activity function was the most enriched (*p* < 0.001, hypergeometric test). The KEGG analysis (Figure 4B) showed that the most enriched pathways and, in turn, those with the highest probability were homologous recombination and non-homologous end-joining. Other pathways such as longevity regulation, meiosis, and mismatch repair were also highlighted (*p* < 0.05, hypergeometric test). It is important to mention that this analysis showed only the involvement of 18 genes (Figure 4C).

Figure 5 shows the GO enrichment analysis obtained for the four common genes found between the phenomenon of resistance/response to radiation and oxidative stress, free radicals, ROS, and antioxidant activity. It can be seen how the biological processes with the highest statistical probabilities were base-excision repair and DNA repair. The most probable and enriched cellular components were the TRC complex and the protein serine/threonine phosphatase complex. Among the most significant and enriched molecular functions, the oxidized purine nucleobase lesion DNA N-glycosilase activity and the class I DNA endonuclease activity stood out (Figure 5A). KEGG analysis showed only the base-excision repair pathway as statistically significant (Figure 5B). A value of *p* < 0.05, from a hypergeometric test, was taken as the significance threshold. After studying the genes obtained, it was found that only *NTG1* and *OGG1* were involved in the KEGG pathway obtained (Figure 5C).

Figure 6 shows the GO analysis performed for the only common gene found among all the terms used in the data mining (*HSP104* gene). It was observed how this gene presented a very high enrichment with a high statistical significance in the trehalose metabolism in response to heat stress, in cellular heat acclimation, and in protein unfolding (Figure 6A). It is part of the TRC complex and nuclear periphery, highlighting ADP binding, chaperone binding, and unfolded protein binding as its main molecular functions. The only KEGG pathway obtained was longevity regulation (Figure 6B). In all cases, the statistical threshold was *p* < 0.05 in a hypergeometric test, (Figure 6C).

The enrichment results suggest that HR and NHEJ are the most important common processes between radiation response/resistance and aging, necessary in maintaining cellular homeostasis. Base excision repair is the main response mechanism to the oxidative stress, free radicals, ROS, and antioxidant activity caused by ionizing radiation. The metabolic routes of longevity regulation are those that jointly contribute to the three phenomena studied.

### 3.4. Enriched Ontology Clusters

The study of enriched ontology clusters performed with Metascape for cluster identification from the 18 genes obtained from the KEGG pathway analysis showed that 12 clusters were involved, standing out from highest to lowest statistical probability as follows: double-strand break repair, mitotic recombination, cell cycle process, gene conversion, non-homologous end-joining, DNA recombinase assembly, the regulation of DNA replication, DNA replication, the negative regulation of DNA metabolic process, the regulation of DNA recombination, longevity regulation pathway, and the response to abiotic stimulus (Figure 7A). Clusters with the highest *p* value were double-strand break repair and mitotic recombination (*p* < 1 × 10^−20^; hypergeometric test) and cell cycle process (*p* < 1 × 10^−10^; Hypergeometric test) (Figure 7B). Analysis with the MCODE algorithm applied to this network allowed for the identification of neighborhoods where proteins are densely connected. A single MCODE cluster (Figure 7C) formed by 11 genes was obtained: *EXO1*, *MEC1*, *MRE11*, *RAD27*, *RAD50*, *RAD51*, *RAD52*, *RAD55*, *RAD9*, *SGS1*, and *YKU70*; (*p* < 0.05; Hypergeometric test) (Figure 7D). The GO enrichment analysis applied to the MCODE network to extract “biological meanings” from the network component showed the top three best *p*-value terms: double-strand break repair (*p* = 5.012 × 10^−16^), DNA repair (*p* = 5.012 × 10^−15^), and cellular response to DNA damage stimulus (*p* = 1.995 × 10^−14^).

Due to the low number of common genes found between radiation response and antioxidant defense, and between all three phenomena, Metascape did not show any clusters between them. The genes identified with MCODE are strongly related to resistance/response to radiation and cellular aging. 

### 3.5. Protein–Protein Interaction Analysis

Stage 3 of this study (Figure 1) consisted in the analysis of protein–protein interactions and the regulatory mechanisms of the eleven genes selected during the enrichment process, identified in the MCODE cluster and the two common genes obtained in Figure 5C.

The interaction analysis resulted in the categorization of the proteins into various groups based on local network clusters (CL, STRING) (Figure 8A,B), KEGG pathways (Figure 8C,D), and Reactome pathways (Figure 8E,F). Edges represent protein–protein associations. Each node represents all the proteins produced by a single protein-coding gene locus. The minimum interaction score (edge confidence) was adjusted to 0.400. In the physical subnetworks the edges indicate that the proteins are part of a physical complex. Line thickness indicate the strength of data support (confidence). The strength values represent how large the enrichment effect is. False discovery rate (FDR) indicates how significant the enrichment is.

The MCODE cluster analysed showed that its proteins were involved in other STRING clusters. The interactions between EXO1, SGS1, MEC1, RAD51, and RAD52, as well as between MRE11, RAD9, and RAD50, stand out with greater strength (Figure 8A). They were mainly involved in the processes of DNA damage and telomere maintenance via telomere lengthening, and DNA repair and DNA topological change, with an FDR value of 4.26 × 10^−14^ and 1.51 × 10^−13^ (STRING clusters: CL:3082 and CL:3085, respectively). Clusters related to DNA damage, homologous recombination, and DNA damage checkpoint, Six1-Six4 complex, and recombinase activity showed a lower FDR value with a higher strength value (CL:3086, CL:3088, CL:3090 and CL:3112). Finally, the clusters with the greatest strength were the DNA damage-induced protein phosphorylation and DNA damage checkpoint, and the Mre11 complex and DNA ligase IV complex (CL:3134, CL:3214) (Figure 8B). 

The MCODE cluster proteins found with Metascape participate in two KEGG pathways related to DNA repair. The network shows that the SGS1, RAD51, RAD52, RAD55, MRE11, and RAD50 proteins were involved in homologous recombination while RAD27, YKU70, MRE11, and RAD50 were involved in non-homologous end-joining, with a high confidence value (Figure 8C). The FDR values obtained were 1.21 × 10^−10^ and 2.2 × 10^−7^, respectively (Figure 8D).

The Reactome pathways analysis showed the participation of the EXO1, RAD51, RAD52, MRE11, and RAD50 proteins (Figure 8E). Most of the pathways found were related to DNA repair. The DNA double-strand break repair route stands out with a higher strength and FDR value (3.26 × 10^−5^) (Figure 8F).

The STRING analysis of the two genes common to radiation resistance and oxidative/antioxidant response (Figure 8G) showed their involvement in the local STRING cluster relative to DNA endonuclease activity (CL:3232) (FDR value = 0.0018). These proteins are involved in the KEGG pathway base-excision repair (FDR = 0.00096) (Figure 8H).

### 3.6. Regulation Mechanisms

After studying the regulation mechanisms of the genes included in the MCODE cluster and the common genes *OGG1*, *NTG1*, and *HSP104*; 52 regulators were obtained (Figure 9). It is worth noting the *RAD50* and *RAD55* genes had only one and two regulators, respectively (Figure 9E,H), followed by *EXO1*, *MRE11*, *RAD27*, *OGG1*, and *NTG1* (Figure 9A,C,D,L,M) that showed three regulators. In contrast, the *MEC1*, *RAD51*, *RAD52*, *RAD9*, *SGS1*, and *YKU70* genes presented four, twelve, nine, six, six, and eight regulators, respectively (Figure 9B,F,G,I–K). It is important to highlight the *HSP104* gene that showed 31 regulators (Figure 9N). Most of these regulators are transcription factors that interact directly with the gene they regulate, through a target molecule or in combination with another regulator. Table 2 shows, in detail, the regulatory molecules of these genes. Almost 50% of the regulators participated in the regulation of transcription in response to heat, while the rest perform it in response to hydrogen peroxide, to the presence of boron-containing molecules, to DNA-damage stimulus, to zinc ion starvation, or to RNA stability.

### 3.7. Human, Mouse, and Rat Homologous Genes and Their Comparison with More Specific Databases

After searching for human, mouse, and rat genes homologous to the common genes found in *S. cerevisiae*, it was found that *RAD55*, *RAD9,* and *NTG1* genes did not display mammalian homologous genes. For the rest of the genes, at least one homologue was obtained (Table 3 and Table 4). Homologous human genes for the phenomena of response/resistance to ionizing radiation and aging are involved in DNA repair through several mechanisms (mismatch repair, nucleotide excision repair, HR, and NHEJ) and to ensure genome stability. Regarding the two common genes of response/resistance to radiation and oxidative stress, free radicals, ROS, and antioxidant activity, only the yeast *OGG1* gene showed a human homologous gene of the same name. In this case, it is a DNA glycosylase that eliminates mutated 8-oxoguanine as a consequence of oxidative damage. The *HSP104* gene proposed for all events showed the CLPB gene as a human homologue. It is involved in several processes, including DNA replication, protein degradation, and the reactivation of misfolded proteins (Table 3).

All mouse and rat homologous genes were similar to those described in humans. Specifically, the genes for the phenomenon of response to radiation and aging are involved in DNA repair mechanisms. These mouse and rat genes participate in various mechanisms that facilitate DSB repair. The Ogg1 gene, with glycosylase activity, is also present in these rodents as a common gene between response to radiation and antioxidant activity. Finally, the *S. cerevisiae HSP104* gene, common to all the phenomena studied, presented the homologous Clpb gene in rodents. The activity of its protein, peptidase in *Mus musculus* and disaggregase in *Rattus norvegicus*, is predicted to be involved in cellular response to heat and in antiviral innate immune response (Table 4).

The comparison of human, mouse, and rat genes with the RadBioBase database (http://radbiodb.physics.ntua.gr/) for ionizing radiation assays showed the results obtained by other authors only for EXO1, RAD51, FEN1, and CLPB genes (Table 5). The obtained studies were performed only in normal human and mouse cells (cardiomyocytes, brain cells, and peripheral blood cells) irradiated mainly with X-rays.

As shown in Table 5, EXO1, RAD51, and FEN1 human genes were down-regulated after 24 and 48 h post-exposure to X-rays (2 and 5 Gy) at a mean dose rate of 1.0 and 2.015 Gy/min; obtaining 12.78 DSB per Gy per Gbp. However, the mouse Exo1, Rad51, and Clpb genes were up-regulated after 0 h, 24 h, and 7 days post-exposure to 2 and 4 Gy with a dose rate of 1.23 Gy/min, to obtain the same effect of DSB.

MGI database showed 390 genes and 583 GO annotations in relation to response to oxidative stress in mouse. These results can be found at the link: https://www.informatics.jax.org/vocab/gene_ontology/GO:0006979 (accessed on 22 August 2023), and the list of genes at: https://www.informatics.jax.org/go/term/GO:0006979 (accessed on 22 August 2023). The mouse homologous genes described in Table 4, obtained from the study in *S. cerevisiae*, are not represented in this list with the exception of Rad52, Wrn, and Ogg1. This suggests that these genes could be the main mouse genes involved in the response to oxidative stress produced by ionizing radiation.

## 4. Discussion

For many years, there has been speculation that there is an important relationship between resistance to radiation and cellular aging. Such a relationship could change the course of research and the treatment of tumors with radiotherapy [7].

The molecular mechanisms of response to physical and chemical agents that cause cell damage are conserved, presenting on numerous occasions the molecular pathways involved in the repair of damage of multiple natures to restore cell homeostasis (genomics and proteomics), maintain cell integrity, and ensure survival with maximum energy savings [6].

Model organisms, such as *S. cerevisiae*, and the use of bioinformatics tools allow us to compare well-known and established molecular mechanisms from curated databases. This methodology provides very valuable information, without the need for experimental laboratory work, in the search for common intra- and inter-species response mechanisms and molecular markers. In this way, the proposed work represents an advance in the subsequent definition, through experimental testing, of the utility of said markers.

Currently, research at the molecular level is guided based on the information obtained in the literature and especially in multiple molecular databases. The methodology used in this work allows obtaining new data from crossing information in curated databases. The works carried out in silico allow use to obtain very valuable information in a very short time and with very few resources. This information, obtained from crossing data in different curated databases, allows us to channel the research work and in some cases, as in this one, discover molecular markers common to various phenomena, which would go undetected in in vitro studies. The data that appear in the databases have been duly validated and accepted for inclusion. This fact reinforces a priori the validity of the markers found. Even so, it is important to carry out new validations of the markers under different exposure conditions. However, the objective of this work is not this but to provide new evidence for further studies. The validation of the proposed markers is left for later studies, where different conditions of exposure to radiation, aging, and response to oxidizing agents are tested. The present work saves time and effort in the search for common markers from data that have already been duly validated for their inclusion in the used curated databases.

Starting from the idea that each phenomenon (resistance/response to radiation, aging, oxidative stress, free radicals, antioxidant activity, ROS) triggers multiple responses with the participation of multiple biological processes, molecular functions, and metabolic pathways, this work demonstrates that among all of them there are common mechanisms to ensure cell homeostasis. This fact facilitates the definition of common molecular markers of response.

Table 6 shows a description of the genes identified with MCODE, strongly related to resistance/response to radiation and cellular aging. Their functions, as well as their main role and the effect of their inhibition on the lifespan of *S. cerevisiae*, are also described. It is worth noting a minimum lifespan reduction range of 15–30% for the *MRE11* gene and a maximum value of 70% reduction for *RAD50* and *RAD52*. The average lifespan reduction was 50%. Among the main functions involved in both phenomena, depending on the greater participation of genes and the greater effect on lifespan were as follows: DNA repair, genomic stability, replication stress, DNA damage, DNA damage response, strand exchange, meiotic recombination, and the maintenance of genome integrity. It is important to highlight that the *HSP104* gene, common to the three phenomena studied, did present a relationship with cellular aging, observing a 40% reduction in life expectancy.

The genes described in *S. cerevisiae* showed homologous genes in mammalian cells of different species. They constitute groups of genes that are very well conserved during evolution, since they are essential for the maintenance of life. They are mainly responsible for repairing the lesions produced in the DNA by physical and chemical agents, and guaranteeing the stability of the genome and telomeres. Specifically, most of the homologous human genes found are involved in DNA DSB repair mechanisms, followed by genes that provide stability to the genome. It is important to highlight in yeast cells the HSP104 disaggregase protein and its human homologue, CLPB, which are both involved in protein degradation and the reactivation of misfolded proteins in response to stress.

The results obtained from the comparison of common genes show that cell aging has a strong relationship with the response/resistance to ionizing radiation. Although the response to oxidative stress is important in the cell when addressing resistance to radiation, there are more common genes with the aging phenomenon. The existence of one single gene (*HSP104*) common to the three phenomena could indicate the possibility of defining a good common biomarker.

The aging phenotype can be progressed or prevented depending on the effectiveness of the damage repair at the protein level. Since radiation damage occurs mainly at the genomic level, the mechanism common to aging must also be at the genomic level, as it has been observed in the gene set enrichment analysis. However, when the three phenomena are considered together (response to radiation, oxidative stress, and aging), the joint response mechanisms must ensure resistance to oxidative damage caused to proteins. In this sense, the HSP104 protein plays a predominant role [13].

Cellular aging is a multifactorial phenomenon at the organism, tissue, and cellular levels. In this sense, it is logical to think about its interrelation with other processes, such as those described for the effects of radiation obtained from the GO enrichment analysis and protein–protein interactions data. This point of view shows that the response/resistance to radiation is also a multifactorial phenomenon that interacts with phenomena that participate in aging and also in the response to ROS, free radicals, oxidative stress, etc.

The increase in damage to DNA causes an increase in ROS levels, causing damage to numerous molecules, which causes an alteration of their functions [6]. These alterations reveal the activation of response mechanisms that are also activated during cell aging, to prevent protein oxidation. In this way, ROS levels activate processes to maintain both genome and proteome integrity [10].

Exposure to ionizing radiation activates all available cellular machinery to prevent/repair DNA and protein damage [12]. The repair of DNA damage is mainly carried out by HR and NHEJ. The mechanisms that prevent the activity of free radicals and ROS also come into play, hence the found interaction with the common response genes *NTG1*, *SIT4*, *OGG1*, and *HSP104*. The last damage caused is on the proteins, mainly by oxidation. Since both the response/resistance to radiation and cell aging are multifactorial phenomena, it is logical to think that there are common molecular mechanisms at the level of DNA and protein damage repair. This fact could explain the large number of common processes and proteins that interact with each other in both phenomena (Table 1). However, this work shows that all the proteins involved in the response/resistance to radiation are mainly involved in processes that seek cellular homeostasis at the genomic level (DNA repair and genomic stability). The homeostasis of the genome is therefore the common phenomenon between the effect of radiation and aging, against the protein homeostasis that would mainly dominate as the majority mechanism to avoid the phenomenon of aging.

However, if we look for a common mechanism of response to the three phenomena, it seems that, according to the data obtained, guaranteeing the correct activity of the proteome is key to maintaining cell survival and controlling aging by minimizing oxidative damage to proteins. In this sense, it is logical that the HSP104 protein is the only common one obtained involved in various metabolic pathways responsible for maintaining longevity and avoiding misfolded proteins. HSP play an important role in cellular proteostasis, also maintaining the correct folding and stabilization of the protein complexes involved in DNA repair and thus contributing to the effectiveness of response mechanisms against DNA-damaging agents [12]. Krisko and Radman [13] used HSP104-GFP aggregates as a marker of aging in *S. cerevisiae* due to the accumulation of misfolded protein aggregates during the yeast lifespan.

The observed regulatory mechanisms highlight the importance of regulation by transcription factors in response to heat. This type of response mediated by HSP constitutes a common response mechanism against different agents that cause physical stress (misfolded and oxidized proteins). The HSP104 protein presents 31 regulators. This characteristic shows the importance that it has in multiple processes to maintain the homeostasis of the proteome, especially the one in charge of repairing DNA damage, as evidenced by the relationship found in this work. Molon and Zadrag-Tecza [29] treated the protein HSP104 as a marker of yeast aging due to its accumulation with age and with the exposure to thermal and oxidative stress, indicating that it is a necessary protein to maintain normal cell life.

This work focused on the phenomenon of aging. Only genes involved in aging and not in senescence were considered. Discrimination was performed in the gene search process using the GenAge database [30]. This database only provides information on aging genes (both replicative and chronological aging) and not on senescence. In this way, although senescence genes could be obtained in the search process for genes related to radiation response/resistance, they were eliminated as they did not coincide with the respective aging genes obtained in the GenAge database after crossing the data from both phenomena.

The results obtained in this work indicate that the HSP104 protein participates in regulatory pathways related to aging, response/resistance to ionizing radiation, oxidative stress, free radical activity, and the response to ROS, as well as antioxidant activity; therefore, they suggest that it could be a good biomarker of these processes.

This study shows that the proposed markers are similar in human, mouse, and rat due to the homology found in almost all genes found for *S. cerevisiae*.

After comparing with specific databases for ionizing radiation (RadBioBase), a different response to irradiation was observed in human and mouse cells. It is striking that human genes (EXO1, RAD51, and FEN1) after exposure to X-rays were down-regulated while mouse genes (Exo1, Rad51 and Clpb) were up-regulated. Although the data found in RadBioBase regarding the genes of interest in this study are scarce, this could indicate a specific response of the species and, to a lesser extent, are specific to the gene evaluated, as is the case of EXO1 and RAD51 genes whose expression state is opposite in human compared with mouse.

After comparing the mouse homologous genes with the MGI database, only the Rad52, Wrn, and Ogg1 genes were obtained as genes involved in the response to oxidative stress. Since these genes are also involved in the response to radiation, this fact suggests that these genes could be the main mouse genes involved in the response to the oxidative damage produced by ionizing radiation. However, searching for these genes in RadBioBase did not produce any results.

## 5. Conclusions

The results found in this work indicate that the response/resistance to ionizing radiation has a strong relationship with the cellular aging phenomenon. In this sense, the HR and NHEJ DNA repair pathways are the most important common processes necessary to maintain cellular homeostasis. On the other hand, the base excision-repair pathway is the main common response mechanism to oxidative stress, free radicals, ROS, and antioxidant activity caused by ionizing radiation. The metabolic routes of longevity regulation are those that jointly contribute to the three phenomena studied. 

The existence of common genes between the studied phenomena indicate the possibility of defining common potential biomarkers in *S. cerevisiae*. The study carried out proposes 11 common biomarkers for the response/resistance to ionizing radiation and aging (EXO1, MEC1, MRE11, RAD27, RAD50, RAD51, RAD52, RAD55, RAD9, SGS1, YKU70) and two biomarkers for response/resistance to radiation and oxidative stress, free radicals, ROS, and antioxidant activity (NTG1, OGG1). In addition, it is important to highlight that the HSP104 protein could be a good biomarker common to the three phenomena studied. 

The experimental validation of the proposed biomarkers has not been considered in this work because it is outside the context of the proposed objectives. Said validation is therefore left for later studies, where different conditions of exposure to radiation, aging, and the response to oxidizing agents are tested. Further studies could define its potential use.

## Figures and Tables

**Figure 1 antioxidants-12-01690-f001:**
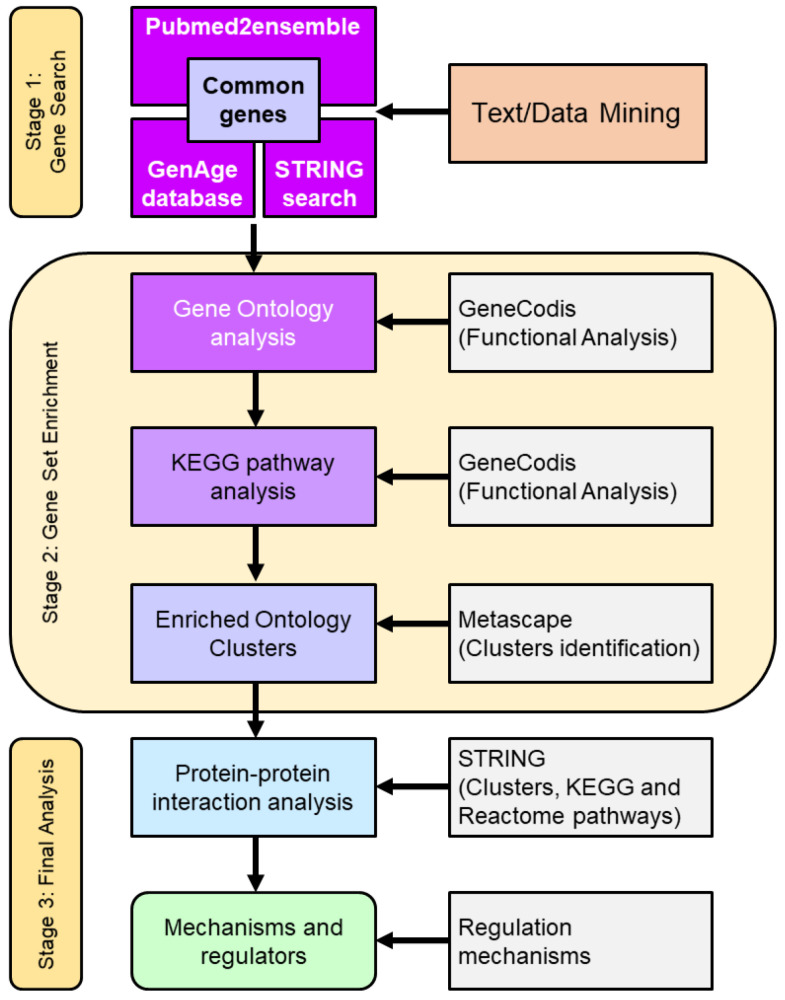
Flowchart used in the search strategy and analysis of genes involved in resistance/response to radiation, aging, oxidative stress, response to free radicals and ROS, and antioxidant activity in *S. cerevisiae*. The study was divided into three stages. Stage 1 corresponded to text/data mining. In this stage, genes associated with the indicated phenomena were identified. Next, stage 2 was called gene set enrichment, and consisted of four functional analyses of the common genes found as well as the identification of clusters. Finally stage 3, or final analysis, in which the main genes involved were obtained and an analysis of protein interactions and regulatory mechanisms was carried out. The number of genes resulting from the text/data mining, the enrichment analysis, and the final selection were reduced according to the statistical significance obtained.

**Figure 2 antioxidants-12-01690-f002:**
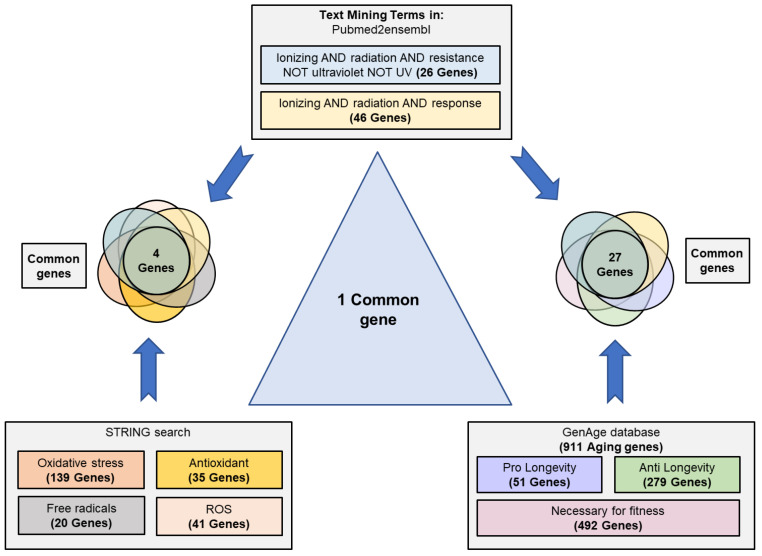
Scheme of text/data mining. Search process and selection of common genes related to the resistance/response to radiation, cellular aging, and their relationship with oxidative stress, free radicals, reactive oxygen species (ROS), and antioxidant activity. Genes related to the concepts “ionizing radiation”, “resistance”, and “response” were identified using Pubmed2ensembl. The terms “oxidative stress”, “free radicals”, reactive oxygen species”, and “antioxidant” were used to search for related genes in STRING. The GenAge database was used to download all genes related to aging in *S. cerevisiae*. The data were crossed to obtain the common genes among all the phenomena.

**Figure 3 antioxidants-12-01690-f003:**
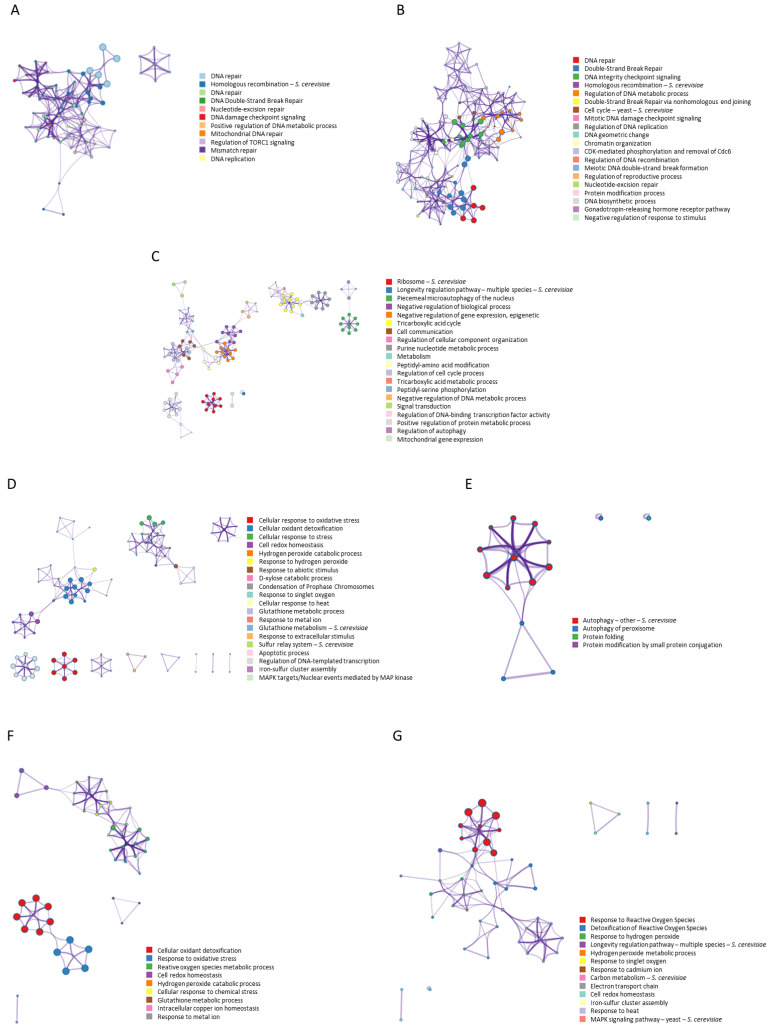
Gene–gene interaction networks and clusters identification for genes related to the search terms: (**A**) “ionizing AND radiation AND resistance NOT ultraviolet NOT UV” (26 genes), (**B**) “ionizing AND radiation AND response” (46 genes), (**C**) 911 aging genes obtained from GenAge database for *S. cerevisiae*, (**D**) 139 genes obtained for “oxidative stress” in STRING, (**E**) 20 genes obtained for “free radicals”, (**F**) 35 genes obtained for “antioxidant”, and (**G**) 41 genes obtained for “reactive oxygen species” (STRING). Cluster networks are colored for cluster identification. Each term is represented by a circle node, where its size is proportional to the number of input genes fall under that term, and its color represent its cluster identity (i.e., nodes of the same color belong to the same cluster). Terms with a similarity score > 0.3 are linked by an edge (the thickness of the edge represents the similarity score). Analysis performed with Metascape.

**Figure 4 antioxidants-12-01690-f004:**
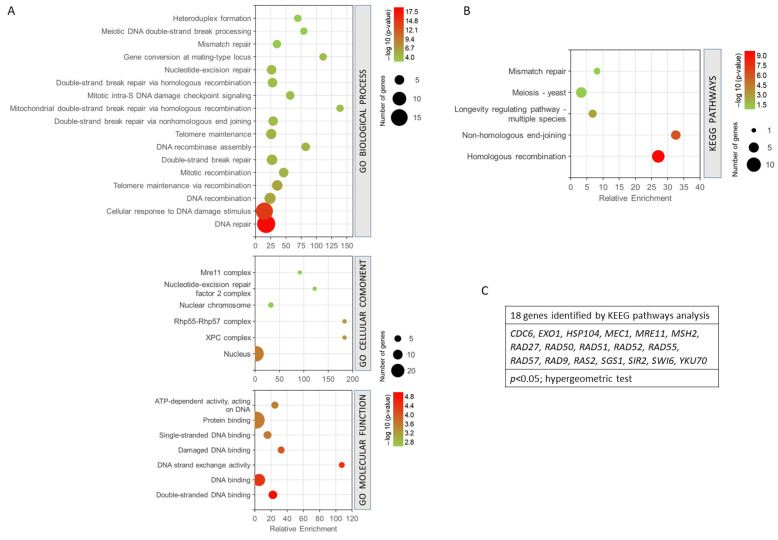
Gene set enrichment analysis for the common genes found for radiation resistance/response and aging. (**A**) GO biological process, *p* < 0.0001; GO cellular component, *p* < 0.005; GO molecular function, *p* < 0.001. (**B**) KEGG pathways, *p* < 0.05. (**C**) Genes identified using KEGG pathways analysis. The analyses were performed using GeneCodis. The hypergeometric test was used as statistical test in all cases.

**Figure 5 antioxidants-12-01690-f005:**
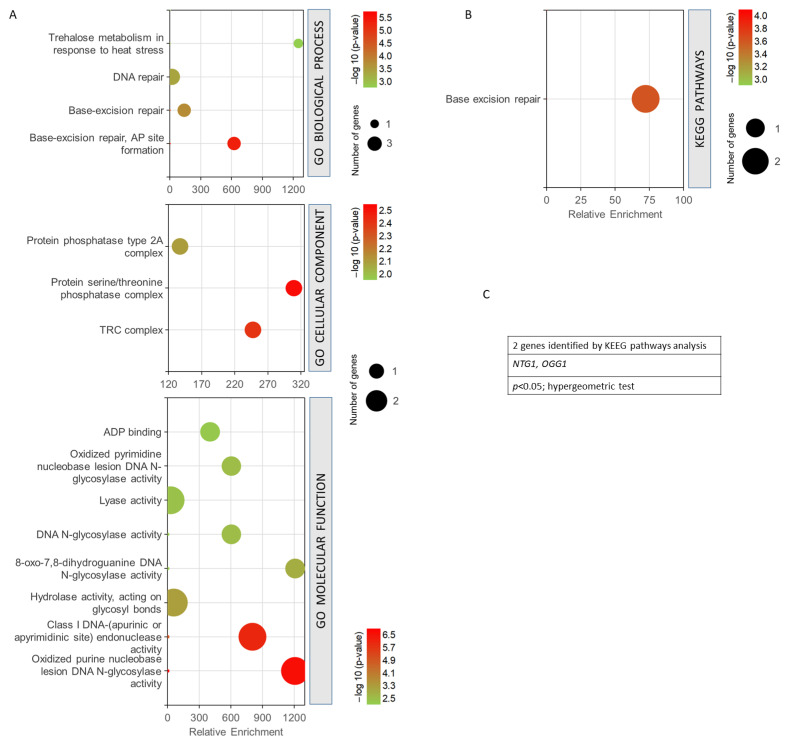
Gene set enrichment analysis for the common genes found for radiation resistance/response and oxidative stress, free radicals, ROS, and antioxidant activity. (**A**) GO biological process, *p* < 0.001; GO cellular component, *p* < 0.05; GO molecular function, *p* < 0.01. (**B**) KEGG pathways, *p* < 0.05. (**C**) Genes identified using KEGG pathways analysis. The analyses were performed using GeneCodis. The hypergeometric test was used as statistical test in all cases.

**Figure 6 antioxidants-12-01690-f006:**
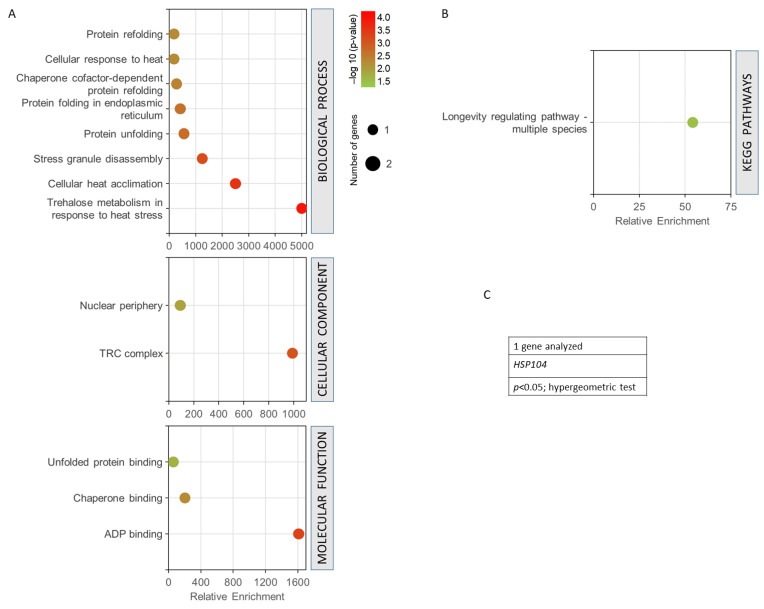
Gene set enrichment analysis for the common genes found for radiation resistance/response, aging, and oxidative stress, free radicals, ROS, and antioxidant activity. (**A**) GO biological process, *p* < 0.01; GO cellular component, *p* < 0.05; GO molecular function, *p* < 0.05. (**B**) KEGG pathways, *p* < 0.05. (**C**) Gen analysed. The analyses were performed using GeneCodis. The hypergeometric test was used as statistical test in all cases.

**Figure 7 antioxidants-12-01690-f007:**
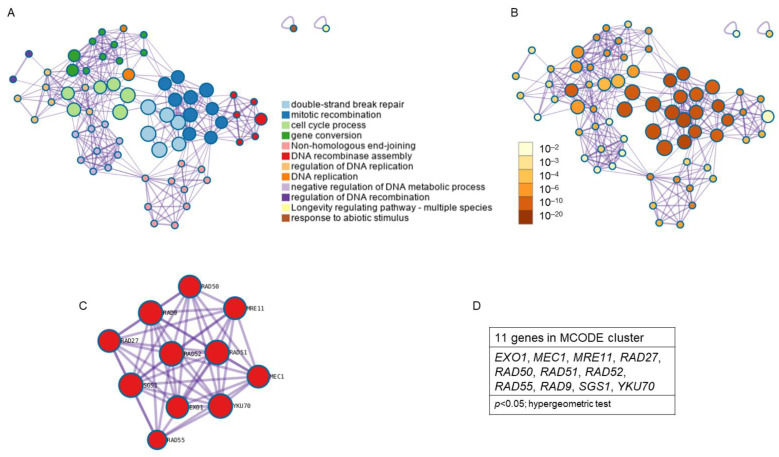
Enriched ontology clusters found for the common genes between resistance/response to radiation and aging. Due to the low number of common genes found between radiation response and antioxidant defense, and between all three phenomena, Metascape did not show any clusters between them. (**A**) Clusters network colored by cluster identification. Each term is represented by a circle node, where its size is proportional to the number of input genes that fall under that term, and its color represents its cluster identity (i.e., nodes of the same color belong to the same cluster). Terms with a similarity score > 0.3 are linked by an edge (the thickness of the edge represents the similarity score). (**B**) Clusters network with nodes colored by *p*-value. (**C**) Protein–protein interaction MCODE components. (**D**) Genes found in the MCODE cluster. *p* < 0.05; Hypergeometric test.

**Figure 8 antioxidants-12-01690-f008:**
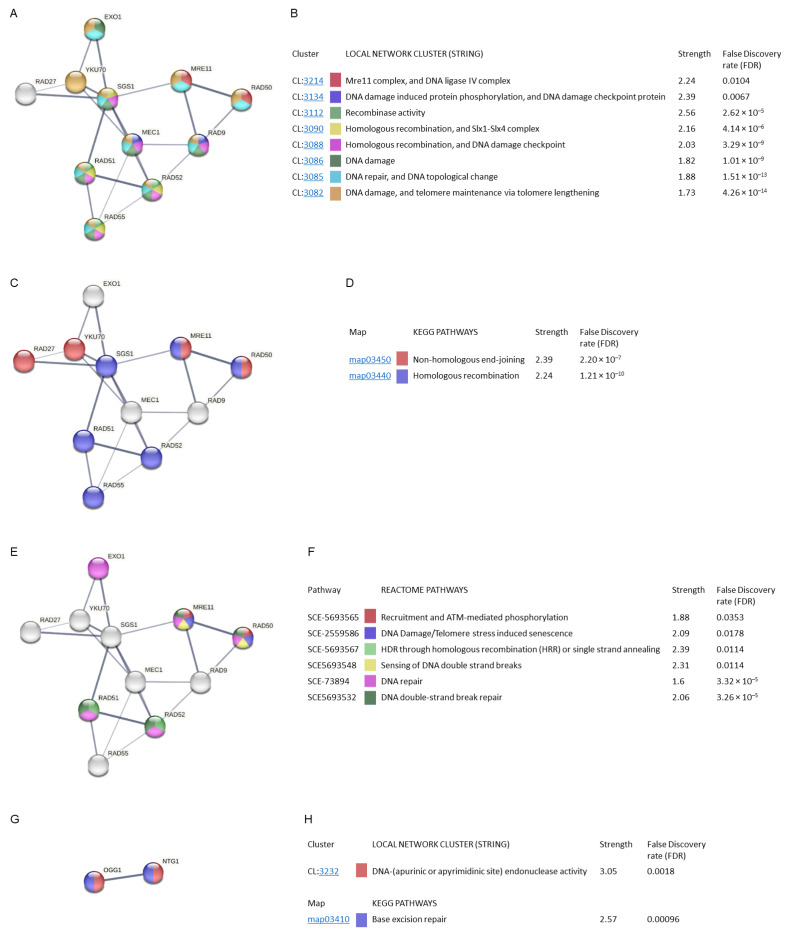
Gene ontology and STRING analysis of protein–protein interactions of MCODE cluster and common proteins involved in radiation resistance, oxidative stress, free radicals, ROS, and antioxidant activity. Edges represent protein–protein associations. Each node represents all the proteins produced by a single protein-coding gene locus. Line thickness indicate the strength of data support. The strength values represent how large the enrichment effect is. False discovery rate (FDR) indicates how significant the enrichment is. (**A**) Local network clusters (CL, STRING). (**B**) Description of local network clusters. (**C**) Network proteins implicated in KEGG pathways. (**D**) Description of KEGG pathways. (**E**) Network proteins implicated in Reactome pathways. (**F**) Description of Reactome pathways. (**G**) Common proteins involved in radiation resistance, oxidative stress, free radicals, ROS, and antioxidant activity. (**H**) Local network clusters (CL, STRING) and description of KEGG pathway.

**Figure 9 antioxidants-12-01690-f009:**
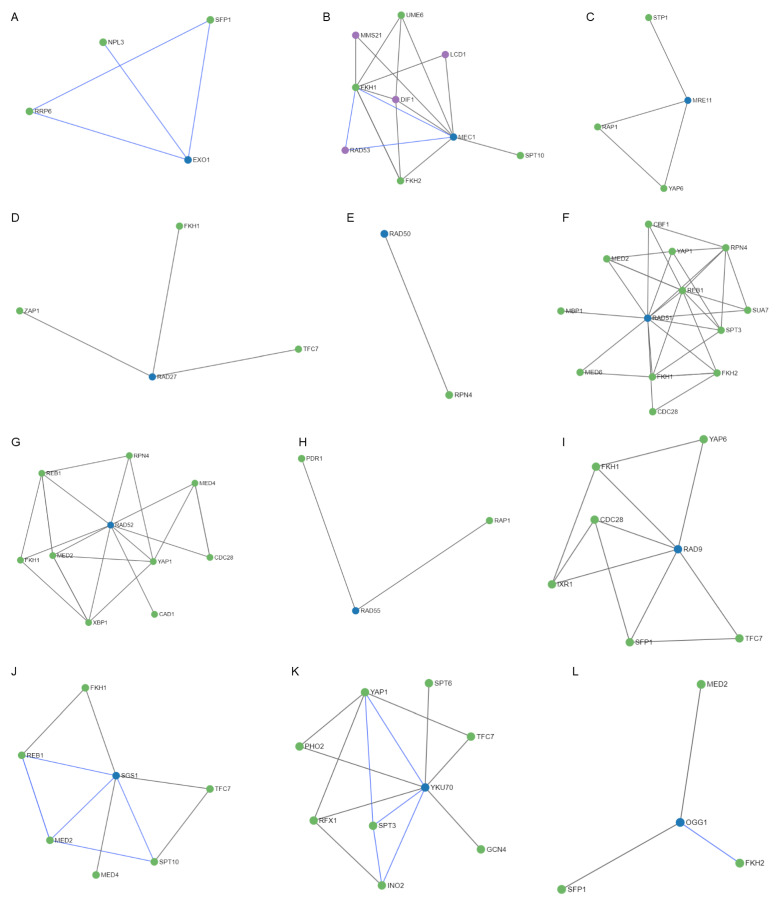
Regulatory networks for the proteins of the MCODE cluster and the common proteins involved in radiation resistance, oxidative stress, free radicals, ROS, antioxidant activity, and aging. (**A**) EXO1. (**B**) MEC1. (**C**) MRE11. (**D**) RAD27. (**E**) RAD50. (**F**) RAD51. (**G**) RAD52. (**H**) RAD55. (**I**) RAD9. (**J**) SGS1. (**K**) YKU70. (**L**) OGG1. (**M**) NTG1. (**N**) HSP104.

**Table 1 antioxidants-12-01690-t001:** Common genes found for the different phenomena after crossing the results obtained from text/data mining.

Radiation Resistance/Response versus Aging	Radiation Resistance/Response versus Oxidative Stress, Free Radicals, ROS and Antioxidant Activity	All Phenomena
*CAT5*, *CDC6*, *DOT1*, *EXO1*, *HSP104*, *MEC1*, *MRE11*, *MSH2*, *NTH1*, *RAD10*, *RAD27*, *RAD34*, *RAD4*, *RAD50*, *RAD51*, *RAD52*, *RAD55*, *RAD57*, *RAD9*, *RAS2*, *RCK2*, *SAS3*, *SGS1*, *SIR2*, *SWI6*, *VPS8*, *YKU70*	*NTG1*, *SIT4*, *OGG1*, *HSP104*	*HSP104*

These genes were common to the terms used in the Pubmed2ensembl search, STRING search, and GenAge database. ROS: Reactive oxygen species.

**Table 2 antioxidants-12-01690-t002:** Molecules that regulate the activity of proteins described in Figure 9.

Regulator	Description	Regulator Type	Regulation of
AFT1	Activator of Ferrous Transport	Transcription factor	Transcription
AFT2	Activator of Fe (iron) Transcription	Transcription factor	Transcription
BUR6	Bypass UAS Requirement	Transcription factor	Transcription in response to heat
CAD1	AP-1-like basic leucine zipper (bZIP) transcriptional activator	Transcription factor	Transcription in response to hydrogen peroxide
CAT8	CATabolite repression	Transcription factor	Transcription
CBF1	Basic helix–loop–helix (bHLH) protein	Transcription factor	Transcription
CDC28	Cyclin-dependent kinase (CDK) catalytic subunit	Protein modifier	Protein activity during G2/M transition of mitotic cell cycle
CUP2	Copper-binding transcription factor	Transcription factor	Transcription
FKH1	Forkhead family transcription factor	Transcription factor	Transcription
FKH2	Forkhead family transcription factor	Transcription factor	Transcription
FLO8	FLOcculation	Transcription factor	Transcription
GCN4	bZIP transcriptional activator of amino acid biosynthetic genes	Transcription factor	Negative transcription in respone to boron-containing substance levels
GCN5	General Control Nonderepressible	Transcription factor	Transcription in response to heat
HSF1	Heat Shock transcription Factor	Transcription factor	Positive Transcription in response to heat
INO2	Transcription factor	Transcription factor	Transcription in response to heat
IXR1	Transcriptional repressor that regulates hypoxic genes during normoxia	Transcription factor	Transcription
MAC1	Metal binding ACtivator	Transcription factor	Transcription
MBP1	Transcription factor	Transcription factor	Transcription
MED2	Subunit of the RNA polymerase II mediator complex	Transcription factor	Transcription in response to heat
MED4	Subunit of the RNA polymerase II mediator complex	Transcription factor	Transcription in response to heat
MED6	Subunit of the RNA polymerase II mediator complex	Transcription factor	Transcription in response to heat
NCB2	Negative Cofactor B	Transcription factor	Transcription in response to heat
NPL3	RNA-binding protein	RNA-binding protein	Positive RNA stability
PDR1	Transcription factor that regulates the pleiotropic drug response	Transcription factor	Transcription
PHO2	Homeobox transcription factor	Transcription factor	Transcription
RAP1	Essential DNA-binding transcription regulator that binds many loci	Transcription factor	Transcription
REB1	RNA polymerase I enhancer binding protein	Transcription factor	Transcription in response to heat
RFX1	Major transcriptional repressor of DNA-damage-regulated genes	Transcription factor	Transcription in response to heat
RGM1	Putative zinc finger DNA binding transcription factor	Transcription factor	Transcription
RGR1	Resistant to Glucose Repression	Transcription factor	Transcription in response to heat
RIM101	Regulator of IME2	Transcription factor	Transcription
RPD3	Reduced Potassium Dependency	Transcription factor	Transcription
RPN4	Transcription factor that stimulates expression of proteasome genes	Transcription factor	Positive transcription during regulation of response to DNA damage stimulus
RRP6	Nuclear exosome exonuclease component	RNA-binding protein	Negative RNA stability
SFP1	Regulates transcription of ribosomal protein and biogenesis genes	Transcription factor	Transcription in response to stress
SPT10	Histone H3 acetylase with a role in transcriptional regulation	Chromatin modifier	Transcription
SPT3	Subunit of the SAGA and SAGA-like transcriptional regulatory complexes	Transcription factor	Transcription in response to heat
SPT6	Histone chaperone	Transcription factor	Transcription in response to heat
SPT7	SuPpressor of Ty’s	Transcription factor	Transcription in response to heat
SRB5	Suppressor of RNA polymerase B	Transcription factor	Transcription in response to heat
STP1	Transcription factor	Transcription factor	Transcription in response to heat
SUA7	Transcription factor TFIIB	Transcription factor	Transcription in response to heat
SWI4	SWItching deficient	Transcription factor	Transcription
TFC7	RNA pol III transcription initiation factor complex (TFIIIC) subunit	Transcription factor	Transcription in response to heat
TUP1	dTMP-UPtake	Transcription factor	Transcription in response to heat
UME6	Rpd3L histone deacetylase complex subunit	Transcription factor	Transcription in response to heat
WTM2	WD repeat containing Transcriptional Modulator	Transcription factor	Transcription
XBP1	Transcriptional repressor	Transcription factor	Transcription in response to heat
YAP1	Basic leucine zipper (bZIP) transcription factor	Transcription factor	Transcription in response to heat
YAP6	Basic leucine zipper (bZIP) transcription factor	Transcription factor	Transcription in response to heat
YHP1	Yeast Homeo-Protein	Transcription factor	Transcription
ZAP1	Zinc-regulated transcription factor	Transcription factor	Positive transcription in response to zinc ion starvation

**Table 3 antioxidants-12-01690-t003:** Human genes homologous to the common genes in *S. cerevisiae*, and their respective function in humans.

Phenomena	*S. cerevisiae* Gene	Homologous Human Gene	Description/Function
Ionizing radiation response/resistance and aging	*EXO1*	EXO1	Exonuclease 1. Involved in mismatch repair and recombination
	*MEC1*	ATR and PRKDC	ATR serine/threonine kinase and protein kinase, DNA-activated, catalytic subunit. ATR can promote DNA repair, recombination, and apoptosis. PRKDC participates in DNA double-strand break repair and recombination
	*MRE11*	MRE11	Double-strand break repair nuclease. Involved in homologous recombination, telomere length maintenance, and DNA double-strand break repair
	*RAD27*	FEN1 and GEN1	Flap structure-specific endonuclease 1 and GEN1 Holliday junction 5’ flap endonuclease. FEN1 removes 5’ overhanging flaps in DNA repair and processes the 5’ ends of Okazaki fragments in lagging strand DNA synthesis. GEN 1 is involved in resolution of Holliday junctions, during homologous recombination and double-strand break repair
	*RAD50*	RAD50	Double-strand break repair protein. Required for nonhomologous joining of DNA ends
	*RAD51*	RAD51	Recombinase. Involved in the homologous recombination and repair of DNA
	*RAD52*	RAD52	RAD52 homolog, DNA repair protein. Important for DNA double-strand break repair and homologous recombination
	*RAD55*	–	–
	*RAD9*	–	–
	*SGS1*	RECQL, RECQL4, WRN, RECQL5, and BLM	Helicases. Involved in DNA repair, including mismatch repair, nucleotide excision repair, and direct repair; in the maintenance of genome stability, replication, transcription, and telomere maintenance
	*YKU70*	XRCC6	X-ray repair cross complementing 6. Involved in the repair of nonhomologous DNA ends
Response/resistance to radiation and oxidative stress, free radicals, ROS, and antioxidant activity	*NTG1*	–	–
	*OGG1*	OGG1	8-oxoguanine DNA glycosylase. Responsible for the excision of 8-oxoguanine, a mutagenic base byproduct which occurs as a result of exposure to reactive oxygen
All phenomena	*HSP104*	CLPB	ClpB family mitochondrial disaggregase. Involved in various processes including DNA replication, protein degradation, and reactivation of misfolded proteins

The description and function of each human gene have been obtained from NCBI (National Center for Biotechnology Information) (https://www.ncbi.nlm.nih.gov/homologene/?term). ROS: Reactive oxygen species.

**Table 4 antioxidants-12-01690-t004:** Mouse (*Mus musculus*) and rat (*Rattus norvegicus*) genes homologous to the common genes in *S. cerevisiae*, and their respective description.

Phenomena	*S. cerevisiae* Gene	Homologous *Mus musculus* Gene	Description of *Mus musculus* Gene	Homologous *Rattus norvegicus* Gene	Description of *Rattus norvegicus* Gene
Ionizing radiation response/resistance and aging	*EXO1*	Exo1	Exonuclease 1	Exo1	Exonuclease 1
	*MEC1*	Atr and Prkdc	Ataxia telangiectasia and Rad3 related; Protein kinase, DNA-activated, catalytic polypeptide	Atr and Prkdc	ATR serine/threonine kinase and protein kinase, DNA-activated, catalytic subunit
	*MRE11*	Mre11a	MRE11A homolog A, double-strand break repair nuclease	Mre11	MRE11 homolog, double-strand break repair nuclease
	*RAD27*	Fen1 and Gen1	Flap structure specific endonuclease 1 and GEN1, Holliday junction 5’ flap endonuclease	Fen1 and Gen1	Flap structure-specific endonuclease 1 and GEN1 Holliday junction 5’ flap endonuclease
	*RAD50*	Rad50	RAD50 double-strand break repair protein	Rad50	RAD50 double-strand break repair protein
	*RAD51*	Rad51	RAD51 recombinase	Rad51	RAD51 recombinase
	*RAD52*	Rad52	RAD52 homolog, DNA repair protein	Rad52	RAD52 homolog, DNA repair protein
	*RAD55*	–	–	–	–
	*RAD9*	–	–	–	–
	*SGS1*	Recql4, Wrn, Recql, Blm, and Recql5	RecQ protein-like 4, Werner syndrome RecQ like helicase, RecQ protein-like, Bloom syndrome, RecQ-like helicase, and RecQ protein-like 5	Recql, Recql5, Wrn, Recql4, and Blm	RecQ-like helicase, RecQ-like helicase 5, WRN RecQ-like helicase, RecQ-like helicase 4, and BLM RecQ-like helicase
	*YKU70*	Xrcc6	X-ray repair complementing defective repair in Chinese hamster cells 6	Xrcc6	X-ray repair cross complementing 6
Response/resistance to radiation and oxidative stress, free radicals, ROS, and antioxidant activity	*NTG1*	–	–	–	–
	*OGG1*	Ogg1	8-oxoguanine DNA-glycosylase 1	Ogg1	8-oxoguanine DNA glycosylase
All phenomena	*HSP104*	Clpb	ClpB caseinolytic peptidase B	Clpb	ClpB family mitochondrial disaggregase

The description and function of each human gene have been obtained from NCBI (National Center for Biotechnology Information) (https://www.ncbi.nlm.nih.gov/homologene/?term). ROS: Reactive oxygen species.

**Table 5 antioxidants-12-01690-t005:** Comparison of human, mouse, and rat genes with the RadBioBase database.

	Human				Mouse		
Gen	EXO1	RAD51		FEN1	Exo1	Rad51	Clpb
Expression status	Down	Down	Down	Down	Up	Up	Up
Tissue/Cell line	iPSC-derived cardiomyocytes	Peripheral blood	iPSC-derived cardiomyocytes	Peripheral blood	Brain	Blood cells	Blood cells
Type of radiation	X-rays	X-rays	X-rays	X-rays	Protons	X-rays	X-rays
Time after irradiation	48 h	24 h	48 h	24 h	0 h	7 days	24 h
LET (keV/μm)	2.1	–	2.1	–	–	2	2
Energy (kV)	240	120	240	120	–	250	250
Dose (Gy)	5	2	5	2	2	4	4
Dose rate (Gy/min.)	1	2.015	1	2.015	–	1.23	1.23
DSB (num./Gy/Gbp)	12.78	12.78	12.78	12.78	–	12.78	12.78

LET: Linear energy transfer.

**Table 6 antioxidants-12-01690-t006:** Potential relevance of proteins in the MCODE cluster and the common protein to all phenomena, to longevity and/or aging.

Gene	Description and Functions	Main Role	Lifespan Effect
*EXO1*	Exodeoxyribonuclease 1; 5′-3′ exonuclease and flap-endonuclease; involved in recombination, double-strand break repair, MMS2 error-free branch of the post replication (PRR) pathway, and DNA mismatch repair; role in telomere maintenance; member of the Rad2p nuclease family, with conserved N and I nuclease domains; relative distribution to the nucleus increases upon DNA replication stress	DNA repair	Reduced
*HSP104*	Chaperone ATPase; Disaggregase; heat shock protein that cooperates with Ydj1p (Hsp40) and Ssa1p (Hsp70) to refold and reactivate previously denatured, aggregated proteins; responsive to stresses including heat, ethanol, and sodium arsenite; involved in [PSI+] propagation; protein becomes more abundant and forms cytoplasmic foci in response to DNA replication stress; potentiated Hsp104p variants decrease TDP-43 proteotoxicity by eliminating its cytoplasmic aggregation; Belongs to the ClpA/ClpB family	Reactivate denatured and aggregated proteins	40% reduced
*MEC1*	Serine/threonine-protein kinase MEC1; Genome integrity checkpoint protein and PI kinase superfamily member; Mec1p and Dun1p function in same pathway to regulate dNTP pools and telomere length; signal transducer required for cell cycle arrest and transcriptional responses to damaged or unreplicated DNA; facilitates replication fork progression and regulates P-body formation under replication stress; promotes interhomolog recombination by phosphorylating Hop1p; associates with shortened, dysfunctional telomeres; belongs to the PI3/PI4-kinase family and the ATM subfamily	Involved in replication stress and DNA damage	66% reduced
*MRE11*	Double-strand break repair protein MRE11; nuclease subunit of the MRX complex with Rad50p and Xrs2p; complex functions in repair of DNA double-strand breaks and in telomere stability; Mre11p associates with Ser/Thr-rich ORFs in premeiotic phase; nuclease activity required for MRX function; widely conserved; forms nuclear foci upon DNA replication stress; belongs to the MRE11/RAD32 family	Meiotic recombination	15–50% reduced
*RAD27*	Flap endonuclease 1; 5′ to 3′ exonuclease, 5′ flap endonuclease; required for Okazaki fragment processing and maturation, for long-patch base-excision repair and large loop repair (LLR), ribonucleotide excision repair; member of the S. pombe RAD2/FEN1 family; relocalizes to the cytosol in response to hypoxia	Genomic stability	50% reduced
*RAD50*	DNA repair protein RAD50; subunit of MRX complex with Mre11p and Xrs2p; complex is involved in processing double-strand DNA breaks in vegetative cells, initiation of meiotic DSBs, telomere maintenance, and nonhomologous end joining; forms nuclear foci upon DNA replication stress; belongs to the SMC family and RAD50 subfamily	Genomic stability	70% reduced
*RAD51*	DNA repair protein RAD51; strand exchange protein; forms a helical filament with DNA that searches for homology; involved in the recombinational repair of double-strand breaks in DNA during vegetative growth and meiosis; homolog of Dmc1p and bacterial RecA protein	Recombination repair	40% reduced
*RAD52*	DNA repair and recombination protein RAD52; protein that stimulates strand exchange; stimulates strand exchange by facilitating Rad51p binding to single-stranded DNA; anneals complementary single-stranded DNA; involved in the repair of double-strand breaks in DNA during vegetative growth and meiosis and UV induced sister chromatid recombination; belongs to the RAD52 family	DNA repair	70% reduced
*RAD55*	Putative DNA-dependent atpase rad55; DNA repair protein RAD55; protein that stimulates strand exchange; stimulates strand exchange by stabilizing the binding of Rad51p to single-stranded DNA; involved in the recombinational repair of double-strand breaks in DNA during vegetative growth and meiosis; forms heterodimer with Rad57p; belongs to the RecA family and RAD55 subfamily	DNA repair	60% reduced/increased
*RAD9*	DNA damage-dependent checkpoint protein; required for cell-cycle arrest in G1/S, intra-S, and G2/M; plays a role in postreplication repair (PRR) pathway; transmits checkpoint signal by activating Rad53p and Chk1p; hyperphosphorylated by Mec1p and Tel1p; multiple cyclin-dependent kinase consensus sites and the C-terminal BRCT domain contribute to DNA damage checkpoint activation; Rad9p Chk1-activating Domain (CAD) is phosphorylated at multiple sites by Cdc28p/Clb2p	Involved in DNA damage response	50% reduced
*SGS1*	ATP-dependent helicase SGS1; RecQ family nucleolar DNA helicase; role in genome integrity maintenance, chromosome synapsis, and meiotic joint molecule/crossover formation; stimulates activity of Top3p; rapidly lost in response to rapamycin in Rrd1p-dependent manner; forms nuclear foci upon DNA replication stress; yeast SGS1 complements mutations in human homolog BLM implicated in Bloom syndrome; also similar to human WRN implicated in Werner syndrome; human BLM and WRN can each complement yeast null mutant; belongs to the helicase family and RecQ subfamily	Involved in maintenance of genome integrity	30% reduced
*YKU70*	ATP-dependent DNA helicase II subunit 1; subunit of the telomeric Ku complex (Yku70p-Yku80p); involved in telomere length maintenance, structure, and telomere position effect; required for localization of telomerase ribonucleoprotein to nucleus via interaction with the TLC1 guide RNA; relocates to sites of double-strand cleavage to promote nonhomologous end joining during DSB repair	Telomere length maintenance	Reduced

The description and function of each gene have been obtained from the STRING database (https://string-db.org/) (accessed on 24 April 2023). The main role and the percentage of lifespan effect have been obtained from the GenAge database (https://genomics.senescence.info/genes/index.html) (accessed on 24 April 2023).

## Data Availability

The data presented in this study are available on request from the corresponding author.

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
