# Peer review of "Molecular Mechanisms of Resistance to Ionizing Radiation in S. cerevisiae and Its Relationship with Aging, Oxidative Stress, and Antioxidant Activity"

_antioxidants, 2023, doi:10.3390/antiox12091690_

Round 1

Reviewer 1 Report

General comments:

This study proposes 11 common biomarkers for response/resistance to ionizing radiation and aging (EXO1, MEC1, MRE11, RAD27, RAD50, RAD51, RAD52, RAD55, RAD9, SGS1, YKU70) and 2 biomarkers for reponse/resistance to radiation and oxidative stress, free radicals, ROS and antioxidant activity (NTG1, OGG1).

Major comments:

1. Table 2: Description and functions may need references.

2. Table 2: The authors must explain how to evaluate the %.

3. Although this is the IR study for yeast, the potential function of these 11 common biomarkers for response/resistance to ionizing radiation and aging must have similar function to mammalian cells. Please discuss these participation of these potential common markers in mammalian cells.

Minor comments:

1. Figure 8 and Figure 9: the size of the gene name is too small to see.

2. p value: p should be typed in italic font.

Author Response

REVIEWER #1

Comments for Authors and responses

General comments:

This study proposes 11 common biomarkers for response/resistance to ionizing radiation and aging (EXO1, MEC1, MRE11, RAD27, RAD50, RAD51, RAD52, RAD55, RAD9, SGS1, YKU70) and 2 biomarkers for response/resistance to radiation and oxidative stress, free radicals, ROS and antioxidant activity (NTG1, OGG1).

Thank you very much for the review and your comments and suggestions to improve the article.

Changes made in the manuscript have been marked in yellow.

Major comments:

  1. Table 2: Description and functions may need references.

The genes in Table 2 are described in the STRING database and their role in aging is described in the GenAge database.

The following text has been included at the bottom of table 2 to clarify the origin of these data and the percentage of lifespan effect:

“The description and function of each gene have been obtained from the STRING database (https://string-db.org/). The main role and the percentage of lifespan effect have been obtained from the GenAge database (https://genomics.senescence.info/genes/index.html)

  1. Table 2: The authors must explain how to evaluate the %.

The expressed percentage has not been evaluated. This percentage appears in the GenAge database as validated data of the inhibition of this gene on lifespan effect. The clarification has been included at the bottom of table 2. Table 2 has changed its number to read “Table 4” because it has been moved to discussion section as suggested by referee 2.

  1. Although this is the IR study for yeast, the potential function of these 11 common biomarkers for response/resistance to ionizing radiation and aging must have similar function to mammalian cells. Please discuss these participation of these potential common markers in mammalian cells.

Thank you very much for this suggestion. Certainly, incorporating the homologous genes to the proposed markers provides new information for future works in human cells.

The subsection "2.8. Human homologous genes" has been included in the Materials and Methods section, where it is described how the search for homologous genes has been done.

In addition, the subsection "3.7. Human homologous genes" has been included in the results section to describe the findings. Homologous genes to all proposed markers have been searched for the three phenomena studied. Homologous genes with their description and function in humans have been compiled in Table 3.

Moreover, the following text has been included in discussion section:

“The genes described in S. cerevisiae showed homologous genes in mammalian cells of different species. They constitute groups of genes that are very well conserved during evolution, since they are essential for the maintenance of life. They are mainly responsible for repairing the lesions produced in the DNA by physical and chemical agents, and guaranteeing the stability of the genome and telomeres. Specifically, most of the homologous human genes found are involved in DNA DSB repair mechanisms, followed by genes that provide stability to the genome. It is important to highlight in yeast cells the HSP104 disaggregase protein and its human homologue CLPB, both involved in protein degradation and reactivation of misfolded proteins in response to stress.”

Minor comments:

  1. Figure 8 and Figure 9: the size of the gene name is too small to see.

The size of figures 8 and 9 has been increased.

  1. p value: p should be typed in italic font.

All “p” has been changed to the italic font. The entire word has been marked in yellow for localization along the text of the manuscript.

Reviewer 2 Report

The major objective of the manuscript presented by González-Vidal et al. was to investigate markers of radiation response/resistance in yeast and their relationship with aging, oxidative stress and antioxidants. In more detail, by using in-silico systems biology approaches, authors indicate that homologous recombination, non-homologous end joining and base excision repair pathways are the most important common pathways of repair. They further propose markers EXO1, MEC1, 30 MRE11, RAD27, RAD50, RAD51, RAD52, RAD55, RAD9, SGS1, YKU70, NTG1, OGG1 to present biomarkers for radiation response, aging and oxidative response, while HSP104 may display a common marker for all of these phenomenons.

While the data, in principle, are interesting, and would warrant further investigations, findings presented are restricted to in-silico readouts that limit the validity of the manuscript. Major shortcomings are given below.

Major points of criticism:

1. The analyses were restricted to standard online bioinformatics tools on publicly available common datasets. To the reviewer´s point of view, publication in a higher-ranking journal requires an experimental validation of at least some of the markers, e.g. HSP104 to ensure correctness of the findings.

2.   Introduction section. Especially the first paragraph covers to a huge extent statements that are common knowledge in radiation biology for decades and thus, should be shortened.

3. Authors searched for genes associated with ionizing radiation and resistance. However, which criteria were used to define radiation resistance? Authors should more clearly define this point.

4.  Gene sets for the biological process ionizing radiation, aging and oxidative response investigated were obtained from three different publicly available datasets. Authors should clearly indicate objectives for using this strategy.

5. Further, table 2 covers published functions of the genes of the MCODE cluster, but does not include any own findings and may better fit to the discussion section.

6. Authors focused on the phenomenon aging. How did they discriminate their findings from cellular senescence, a well-described mechanism following exposure to ionizing radiation?

Author Response

REVIEWER #2

Comments for Authors and responses

The major objective of the manuscript presented by González-Vidal et al. was to investigate markers of radiation response/resistance in yeast and their relationship with aging, oxidative stress and antioxidants. In more detail, by using in-silico systems biology approaches, authors indicate that homologous recombination, non-homologous end joining and base excision repair pathways are the most important common pathways of repair. They further propose markers EXO1, MEC1, 30 MRE11, RAD27, RAD50, RAD51, RAD52, RAD55, RAD9, SGS1, YKU70, NTG1, OGG1 to present biomarkers for radiation response, aging and oxidative response, while HSP104 may display a common marker for all of these phenomenons.

While the data, in principle, are interesting, and would warrant further investigations, findings presented are restricted to in-silico readouts that limit the validity of the manuscript. Major shortcomings are given below.

Thank you very much for the review and your comments and suggestions to improve the article.

Changes made in the manuscript have been marked in green.

Major points of criticism:

  1. The analyses were restricted to standard online bioinformatics tools on publicly available common datasets. To the reviewer´s point of view, publication in a higher-ranking journal requires an experimental validation of at least some of the markers, e.g. HSP104 to ensure correctness of the findings.

Thank you very much for this comment and suggestion to improve the manuscript.

It is true that the data obtained in-silico must be validated experimentally due to the wide range of experimental conditions of assay. Regardless of the rank of the journal, the works carried out in-silico allow obtaining very valuable information in a very short time and with very few resources. This information, obtained from crossing data in different databases, allows us to channel the research work and in some cases, as in this one, discover molecular markers common to various phenomena; that in in vitro studies would go undetected. The data that appears in the databases have been duly validated and accepted for inclusion. This fact reinforces a priori the validity of the markers found. Even so, it is true that it is important to carry out a new validation of the marker under different exposure conditions. However, this is not the objective of this article; but to provide new evidence for further studies. Validation of a single marker requires testing under multiple conditions of both radiation exposure and response to oxidizing agents. This would greatly multiply the results in an article that is already long. For this reason, this validation is left for later studies, where different conditions of exposure to radiation, aging and response to oxidizing agents are tested. The present work saves time and effort in the search for common markers from data that have already been duly validated for their inclusion in the curated databases.

In this way, the following text (marked in green in discussion): “Model organisms, such as S. cerevisiae, and the use of …, of the utility of said markers.”, explain the advantage of in-silico studies.

The following text has been added to discussion to clarify the strategy of in-silico studies and the validity of the data used:

“Currently, research at the molecular level is guided based on the information obtained in the literature and especially in multiple molecular databases. The methodology used in this work allows obtaining new data from crossing information in curated databases. The works carried out in-silico allow obtaining very valuable information in a very short time and with very few resources. This information, obtained from crossing data in different curated databases, allows us to channel the research work and in some cases, as in this one, discover molecular markers common to various phenomena; that in in vitro studies would go undetected. The data that appears in the databases have been duly validated and accepted for inclusion. This fact reinforces a priori the validity of the markers found. Even so, it is important to carry out new validations of the markers under different exposure conditions. However, this is not the objective of this work; but to provide new evidence for further studies. Validation of the proposed markers is left for later studies, where different conditions of exposure to radiation, aging and response to oxidizing agents are tested. The present work saves time and effort in the search for common markers from data that have already been duly validated for their inclusion in the used curated databases.”

In addition, some authors have proposed the HSP104 protein as a marker of aging. This fact has been indicated in discussion section with the following text (marked in green): “Krisko and Radman [13] used HSP104-GFP aggregates as a marker of aging in S. cerevisiae due to the accumulation of misfolded protein aggregates during the yeast lifespan.”

Moreover, other authors found that this protein accumulates with age and with exposure to thermal and oxidative stress. This fact is indicated in discussion with the text (marked in green): “Molon and Zadrag-Tecza [29] treated the protein HSP104 as a marker of yeast aging due to its accumulation with age and with the exposure to thermal and oxidative stress, indicating that it is a necessary protein to maintain normal cell life.

Therefore, we understand, given that curated databases show experimentally validated information and, furthermore, numerous authors have found a direct relationship between the level of expression of the HSP104 protein with aging, oxidative stress and the response to heat; that it is very plausible to assume that the HSP104 protein could be a good biomarker candidate without the need to carry out a new validation in the context of this work. Therefore, a new validation is justified in any case for subsequent studies where different exposure conditions are tested. In this way, the following text has been included at the end of the conclusion section:

The experimental validation of the proposed biomarkers has not been considered in this work because it is outside the context of the proposed objectives. Said validation is therefore left for later studies, where different conditions of exposure to radiation, aging and response to oxidizing agents are tested.”

In this way, it is clear and without a doubt what the objectives of the work are, the scope and advantages of using the curated databases, the validity of their data and the need for new validations in diverse experimental situations to be defined in new studies. These clarifications avoid doubts and misconceptions among readers.

  1. Introduction section. Especially the first paragraph covers to a huge extent statements that are common knowledge in radiation biology for decades and thus, should be shortened.

The first paragraph of the introduction has been shortened to read:

“Ionizing radiation produces different types of DNA damage, with double-strand breaks (DSB) being especially important. In this sense, the repair capacity contributes not only to the maintenance of the genome integrity but also to the resistance to radiation [1-3]. In addition, the response to radiation to ensure cell homeostasis activates other mechanisms involved in cell cycle blockage, free radicals’ formation and apoptosis inhibition [1].”

  1. Authors searched for genes associated with ionizing radiation and resistance. However, which criteria were used to define radiation resistance? Authors should more clearly define this point.

Since there is no specific database of genes/proteins for resistance or response to ionizing radiation, the search for them was carried out with the search engine pubmed2ensembl.

To address this point and clarify the criteria used to define radiation resistance, the following texts were added to section 2.2. Text/data mining:

“Since there is no specific database of genes/proteins for resistance or response to ionizing radiation, the text mining for them was carried out…”

“Numerous authors have described the genes that participate in the response to radiation, also referring to the fact that they participate in resistance phenomena. The criteria used to define radiation resistance were the participation of genes/proteins in processes described by the authors that are correlated with:

  • DNA repair mechanisms (DSB repair, HR, NHEJ, base excision repair, mismatch repair, etc.).
  • DNA-damage checkpoint control proteins (mitosis entry checkpoint, telomere length regulation, etc.).
  • Cell cycle division control (G1/S-specific cyclins, cell-cycle box factors and regulatory proteins in response to radiation).
  • Heat shock response.
  • Proteins that relate cell proliferation with resistance to radiation.
  • Regulatory proteins and post-replication repair ubiquitin-proteins in response to radiation.
  • General transcription and DNA repair factors.
  • Other processes related to radiation response (proteins involved in joint resistance to radiation and metals and/or drugs, helicases, and transcriptional coactivators).”

  1. Gene sets for the biological process ionizing radiation, aging and oxidative response investigated were obtained from three different publicly available datasets. Authors should clearly indicate objectives for using this strategy.

The search strategy used for each process has been different for several reasons. The most complete and updated data source has been used for each biological process.

The text “…radiation, cellular aging and oxidative response was…” has been included in section 2.1. Study strategy and analysis, to clarify.

In this sense, the GenAge database, specific to aging, has been used to search for aging genes. This database contains all validated genes related to cell aging in S. cerevisiae. Therefore, the use of this database makes it possible to discriminate the genes related to the aging phenomenon with respect to those involved in cellular senescence.

The first paragraph of section 2.2. Text/data mining, has been modified to read: “The most complete and updated data source was used to search for specific genes of each biological process. Data mining in the specific GenAge database (https://genomics..../genes/index.html) allowed to search for all validated genes of Saccharomyces cerevisiae involved in the aging process. This database does not contain senescence genes, which makes it possible to discriminate the genes related to the phenomenon of aging with respect to those involved in cellular senescence.”

The search for genes involved in the oxidative response did not show results using the pubmed2ensembl search engine, so we used the STRING database/search engine, which allowed us to obtain results from both the literature and curated databases.

The second paragraph of section 2.2. Text/data mining, has been modified to read: “The search for genes involved in the oxidative response did not show results using the pubmed2ensembl search engine. Therefore, the Search Tool for the Retrieval of Interacting Genes (STRING) database (https://string-db.org/) (11.5 version), was used to identify genes related to oxidative stress, free radicals, antioxidant activity and ROS. This database/search engine, allowed us to obtain results from both the literature and curated databases.

The genes related to resistance/response to radiation have been obtained using the search engine pubmed2ensembl, which allows links between the literature and genes related to different processes, since there is no specific database of genes/proteins involved in resistance. or response to ionizing radiation.

The third paragraph of section 2.2. Text/data mining, has been modified to read: “Since there is no specific database of genes/proteins for resistance or response to ionizing radiation, the text mining for them was carried out using the pubmed2ensembl search engine…

  1. Further, table 2 covers published functions of the genes of the MCODE cluster, but does not include any own findings and may better fit to the discussion section.

Thanks for this suggestion. The text related to table 2 and table 2 itself have been included in the discussion section as suggested. This change in location needs to change the number of the table. Table 2 has changed to read “Table 4” and Table 3 to read “Table 2”. A new Table 3 has been added as suggested by referee 1.

The following text has been moved to discussion section:

“Table 4 shows a description of the genes identified with MCODE, strongly related to resistance/response … with cellular aging, observing a 40 % reduction in life expectancy.”

  1. Authors focused on the phenomenon aging. How did they discriminate their findings from cellular senescence, a well-described mechanism following exposure to ionizing radiation?

Thank you very much for pointing out this important aspect that we had forgotten.

Indeed, the work has focused on aging. The discrimination of genes related to the phenomenon of aging with respect to cellular senescence was carried out in the gene search process using the GenAge database. This database only provides information on aging genes (both chronological and replicative aging) and therefore does not provide information on the genes involved in cell senescence.

This fact is another of the reasons why aging genes were searched in this database and not from literature search engines.

The following text has been added to the first paragraph of section 2.2. Text/data mining; to read: “This database does not contain senescence genes, which makes it possible to discriminate the genes related to the phenomenon of aging with respect to those involved in cellular senescence.

Due to the importance in the discrimination of the phenomenon of aging with respect to that of senescence, the following paragraph has been included at the end of the discussion:

“This work focused on the phenomenon of aging. Only genes involved in aging and not in senescence were considered. Discrimination was performed in the gene search process using the GenAge database [30]. This database only provides information on aging genes (both replicative and chronological aging) and not on senescence. In this way, although senescence genes could be obtained in the search process for genes related to radiation response/resistance, they were eliminated as they did not coincide with the respective aging genes obtained in the GenAge database after crossing the data from both phenomena.

Reference number 30 (Tacutu et al 2018) has been included in the reference list.

Round 2

Reviewer 2 Report

Authors put high efforts in revising the manuscript and my previous concerns were addressed in an adequate and convincing manner.

Author Response

Response to reviewer 2:

Thank you very much for your review and your suggestions. Modifications made based on your comments and suggestions have greatly improved the article. Thank you very much for your time.